# Novel Magnesium- and Silver-Loaded Dressing Promotes Tissue Regeneration in Cutaneous Wounds

**DOI:** 10.3390/ijms25179311

**Published:** 2024-08-28

**Authors:** Shin-Chen Pan, Ying-Jhen Huang, Chong-Han Wang, Chao-Kai Hsu, Ming-Long Yeh

**Affiliations:** 1Department of Surgery, Section of Plastic and Reconstructive Surgery, National Cheng Kung University Hospital, College of Medicine, National Cheng Kung University, Tainan 704, Taiwan; pansc@mail.ncku.edu.tw; 2Department of Biomedical Engineering, National Cheng Kung University, Tainan 701, Taiwan; immortal128125@hotmail.com (Y.-J.H.); deutscnland@gmail.com (C.-H.W.); 3Department of Dermatology, National Cheng Kung University Hospital, College of Medicine, National Cheng Kung University, Tainan 704, Taiwan; kylehsu.tw@gmail.com; 4Medical Device Innovation Center, National Cheng Kung University, Tainan 701, Taiwan

**Keywords:** magnesium, silver, wound dressing, wound healing

## Abstract

Wound healing is a dynamic process involving a complex interaction between many cells and mediators. Magnesium (Mg) is an essential element for cell stabilization. Mg was reported to stimulate the proliferation and migration of endothelial cells in angiogenesis in vitro. However, the function of Mg in wound healing is not known. We observed that the expression level of Mg in human wound tissue fluid was only 10% of that found in human blood serum. To confirm whether Mg is a suitable wound dressing material, we fabricated a Mg- or Mg-silver (Ag)-based polyethylene dressing to study its effect on wound healing. We observed that Mg and Ag were stably preserved in the constructed material and were able to be rapidly released in the moist environment. We also observed that the Mg-based dressing had good cellular compatibility without harmful extractables. Furthermore, Mg enhanced the antibacterial activity of Ag. In line with the observed increase in fibroblast migration in vitro, the Mg-Ag-based dressing improved acute and chronic wound repairs via an increase in neovascularization and basal cell proliferation. The present results show that a Mg-Ag-based coating can be manufactured as an optimal dressing for adjuvant wound therapy.

## 1. Introduction

Wound healing is a dynamic process involving a complex interaction between many cells and mediators [1]. Effective wound healing reduces the risk of complications such as infections, which can lead to more severe health issues, including sepsis, amputation, or even death. Chronic wounds, if left untreated, can result in significant morbidity and complications that place a burden on healthcare systems and patients alike. It has been shown that this interaction affects several important processes in wound healing, such as re-epithelialization, neovascularization, and connective tissue deposition [2]. Most studies emphasized the functions or signaling of cells and mediators to perform regulatory events. However, the significance of trace elements in wound healing warrants further investigation. The role of zinc in wound healing has been established. High levels of zinc and its binding protein, metallothionein, are expressed in the keratinocytes at the wound border and other cell types during the early inflammatory phase of the rat wound model [3]. The application of topical zinc oxide enhanced healing in both partial-thickness and full-thickness wounds [4]. A significantly differential level of copper between normal skin and scar tissue suggested the role of trace elements in the regulation of the wound healing process [5].

Magnesium (Mg) is an essential element for cell stabilization. For example, ATP-generating reactions are Mg-dependent [6]. Other functions of Mg include the regulation of muscle activity, control of neurological function, and neurotransmitter release. It also contributes to maintaining the function of the heart, peripheral vessels, platelets, and bone formation [6]. However, the function of Mg in wound healing has not been studied. Mg is mostly found in muscle tissue, soft tissue, and bone [7]. The distribution of Mg in the skin was not addressed until now. Mg was reported to stimulate the proliferation and migration of endothelial cells in angiogenesis in vitro [8]. Low Mg levels enhanced the dysfunction of cultured endothelial cells via the activation of pro-inflammatory cytokines, such as IL-1 [9]. In addition, Ag is a well-known antibacterial agent. A previous report demonstrated that silver nanowires were potential antibacterial agents against *E. coli*, *S. aureus*, MRSA, and *S. saprophyticus* [10]. In addition, silver nanoparticles have been shown to be effective against a broad spectrum of viruses and also possess a potent antibacterial effect [11]. Herein, we aimed to develop novel Mg- and Ag-based dressings for a dual effect of antimicrobial activity and enhanced wound healing rate. We have conducted a detailed characterization of the dressing, including immersion, microstructure, and surface wettability. The cytotoxic and antimicrobial effects of the dressing were also tested.

## 2. Results

### 2.1. Differential Expression of Mg in Human Serum and Wound Tissue

Local healing responses may lead to differences in systemic and local mediator production. To investigate whether Mg levels were different between serum and wound tissue, a total of 17 individuals were included to test the Mg levels in blood and wound tissue fluids. Of interest, we found that only 10% of the Mg expression level was observed in wound tissue fluid (0.24 ± 0.01 mg/dL), compared to expression in blood serum (2.25 ± 0.21 mg/dL, Figure 1), suggesting the potential of local Mg additives to promote wound healing.

### 2.2. Efficient Release Properties of Mg- and Ag-Containing Dressings

The influence of Mg in cell physiology has been well-demonstrated. Given the evidence that an ideal wound dressing should provide efficient release of the components to promote wound healing, reduce bacteria counts, and minimize tissue toxicity [12], a thorough understanding of Mg and Ag release kinetics at different time points was studied using an immersion test from 2 to 72 h. At 6 h, over 50% of the Mg ion was observed in the medium and ascended to its highest value at 72 h (2.19 mM, Appendix A). In addition, two different types of Mg dressings (Mg100 and Ag200 + Mg100) exhibited a similar release property (Appendix A). Furthermore, the release of the Ag ion in the medium was 0.5 ppm at 2 h, 0.89 ppm at 6 h, 0.88 ppm at 24 h, and 0.96 ppm at 72 h. The Ag ion was observed to rapidly release into the medium at 6 h, exhibiting good antibacterial activity, reaching approximately 92% of its concentration at 72 h (Appendix A). 

### 2.3. Incorporation of Mg and Ag in Wound Dressings

Polyethylene materials are widely used in wound dressings. The mesh coated with Ag and Mg ions was prepared through sputtered technology. In the Ag200 + Mg100 dressing, the Mg ions were sputtered on top of the Ag ions to keep the Mg ions on the outermost layer of the dressing. The surface morphology of the sputtered metal ions was observed by scanning electron microscopy (SEM) at 250× and 500× magnification. Either Ag or Mg were welded steadily to maintain the integrity of the material in use (Appendix A). In addition, surface elements such as Ag and Mg ions were studied with energy-dispersive X-ray spectroscopy (EDS) analysis. The data showed that the average proportion of Mg atoms was 4.7% in Mg100 dressing and 8.7% in Ag200 + Mg100 dressing, with Ag atoms at a low proportion of 0.15% in the latter (Appendix A). The results indicated that Mg and Ag ions can be incorporated into our material by a sputtered process.

### 2.4. Good Surface Wettability of Mg100 and Ag200 + Mg100 Dressings

Wettability is one of the important abilities of sputtered technology. Traditionally, the contact angle was utilized to measure the wettability between the test liquid and the solid surface [13]. A contact angle greater than 90 degrees is needed to maintain its hydrophobic property to avoid stickiness of the dressing to the tissue wound, reduce secondary damage during dressing change, and also help to release ions. To study the surface wettability of our materials, the contact angles of different thicknesses of our dressing, such as 10, 25, 50, and 100 nm, were measured. The results showed that the contact angles of all materials were about 120 degrees. Additionally, there were no differences in the contact angle between uncoated dressing and all types of Mg dressings (Appendix A), suggesting the Mg coating did not significantly alter the hydrophobic properties of the dressing material. 

### 2.5. Absence of Harmful Extractables from Mg100 and Ag200 + Mg100 Dressings 

To study the toxic effects of Ag and Mg in our material, a standard cell toxicity test was used. According to the policy of ISO 10993-5 [14], we picked HUVEC and NIH-3T3 cell lines for cytotoxicity testing of our dressings. A maximal Mg release property of 2 mM was determined to prevent cell toxicity in our design. The extracts of Mg100 and Ag200 + Mg100 dressings were diluted into four different conditions for the following experiment. Solutions with 10% DMSO and 10% FBS were indicated as positive and negative controls, respectively. The HUVECs displayed a higher survival rate (>70%) when cultured at different levels of Mg concentrations, ranging from 0.1 to 2 mM (Figure 2A). By contrast, cultured NIH-3T3s still displayed a higher survival rate in media with 0.5 and 1 mM Mg concentrations, which were 94.63% and 78.75%, respectively. However, the survival rate of the NIH-3T3s decreased to 64.92% in the 2 mM Mg medium (Figure 2B). Although decreasing viability of fibroblasts was observed, cell viability could be reversed to higher than 70% in the Ag-Mg combination, indicating a potentially synergistic effect of Ag and Mg in the dressing. In addition, HUVECs and NIH-3T3s exhibited a high survival rate in the medium of different concentrations of Ag and Mg in the Ag200 + Mg100 dressing (Figure 2C,D). The present study showed HUVECs were able to maintain a high viability rate at this level. Even in the presence of 0.8 ppm of Ag with Mg, cell viability still reached 82.39%, indicating the absence of harmful extractables from our Ag- and Mg-based materials.

### 2.6. Magnesium-Enhanced Antibacterial Ability of Silver in Wound Dressing 

Testing for the antimicrobial ability of wound dressings was carried out using *Staphylococcus aureus*, *Klebsiella pneumonia*, *Escherichia coli*, and *Pseudomonas aeruginosa* [15]. According to the laboratory’s Biosafety Level (BSL) guidelines, a BSL-1 *E. coli* strain was chosen to test our products. The data showed that both the Mg100 and Ag200 + Mg100 dressings had lower OD600 values, with the Ag200 + Mg100 dressing showing a significantly lower value (*p* < 0.05, n = 3) compared to the control (Figure 3A). The obtained value was also converted into the efficiency of the bacteriostatic activity. The data indicated that the Ag200 + Mg100 dressing exhibited 40% of bacteriostatic activity against *E. coli* (Figure 3B), suggesting that Mg amplifies the antibacterial properties of Ag.

### 2.7. Increased Migration of Fibroblasts by Mg100 and Ag200 + Mg100 Dressings

Mg was found to promote fibroblast migration and accelerate wound healing in optimal conditions [16]. To study the effect of Ag and Mg on cell migration, we performed an in vitro scratch wound assay. Fibroblasts were incubated in the medium containing either Mg100 or Ag200 + Mg100 dressings for 24 h. Images of the scratch wounds at 6 and 24 h after scratching showed that fibroblast migration into the scratch wound area was accelerated in the presence of Mg (Figure 4A). At 24 h, the scratch wound was nearly closed for the medium containing Mg100 or Ag200 + Mg100, whereas the control medium had no such effect. Quantification of the percentage of scratch wound closure using image analysis confirmed a significant acceleration of fibroblast migration in the presence of Mg100 or Ag200 + Mg100 at 6 and 24 h (Figure 4B). 

### 2.8. Increased Cutaneous Wound Healing in Normal and Diabetic Mice by Mg Dressings

Although Mg has been reported to promote endothelial cell functions [8], its exact role in cutaneous wound repair is unclear. We have demonstrated the promising effects of the Ag200 + Mg100 dressing in in vitro experiments. However, the specific influence of varying Ag levels on wound healing remains unclear. To explore the effects of increased Ag concentrations on wound healing, we have developed two distinct Ag dressing variants (Ag200 + Mg100 and Ag2000 + Mg100) for in vivo studies. We created four 1 cm full-thickness skin wounds on the back of SD rats. To determine the optimal concentration, three different concentrations of Mg dressings, including 100 mg, 200 mg, and 300 mg, were developed. The results show that the three different types of Mg dressings provide similar wound-closure responses (Appendix A). Due to the reports of antimicrobial effects and possible acceleration of wound healing of Ag [17,18], the synergistic effect of Mg and Ag was examined. The punch wounds were covered with Mg100, Ag200 + Mg100 or Ag2000 + Mg100 dressings. Each dressing manifested an improvement in the wound closure from day 1 to day 7, with significant responses of Ag200 + Mg100 and Ag2000 + Mg100 at day 1 and day 3 when compared to the control (Figure 5A,B). The quantification of hematoxylin and eosin (H&E)-stained sections showed a faster healing rate of neo-epidermis of Mg100 and Ag200 + Mg100 wounds at day 9 post-wounding (Figure 5C,D). To examine if the improvement in wound healing was due to the increase in re-epithelialization, we scored the Ki67+ proliferating keratinocytes at day 9 after establishing the wound (Figure 5E). The experimental groups exhibited an increase in Ki67+ keratinocytes in the wound area, with a significant number in Mg100 dressing when compared to the control group (Figure 5F). Neovascularization is an important process of wound healing and precedes re-epithelialization. To further delineate the effect of Mg on the acceleration of in vivo vasculature, we detected and quantified blood vessels with endothelial marker CD31 immunohistochemistry in wound sections (Figure 5G). Examining sections from different wound tissues revealed that the Mg100 dressing showed a greater percentage of vasculature in the dermis (Mg100: 6.4 ± 0.7%, Ag200 + Mg100: 5.5 ± 0.5%, and Ag2000 + Mg100: 6.0 ± 0.8%). By contrast, a significantly lower percentage of CD31+ blood vessels was observed in the control group (3.5 ± 0.5 %, *p* < 0.05, Figure 5H). In addition, we did not observe a statistical significance in the different Ag dressing variants. Based on these observations, we conclude that Mg promotes acute wound healing, partly via increasing basal cell proliferation. 

We also tested the effects of the Ag and Mg dressings in a diabetic mouse model. After successfully inducing hyperglycemia in SD rats (>250 mg/dL blood glucose, Appendix A), four 1 cm full-thickness skin wounds were created using a biopsy punch 7 days later. The healing effects of the dressings (Mg100, Ag200 + Mg100, and Ag2000 + Mg100) on diabetic wounds were studied. The data showed a significant improvement of wound closure rate in the Mg100 and Ag200 + Mg100 groups from day 1 to day 5 (Figure 6A,B). The neo-epidermis of different wounds was further examined by H&E stain of day-9 wound sections. The results showed a better re-epithelialization with the Ag200 + Mg100 dressing (88.2 ± 7.9%) compared to the control (51.4 ± 7.8%, *p* < 0.05, Figure 6C,D). The Ki67+ keratinocytes of day-9 tissues were further investigated (Figure 6E). Despite no significant difference, a trend toward an increase in Ki67+ keratinocytes in wound sections with the Mg100 (5.8 ± 0.8 cells/200 μm) and Ag200 + Mg100 (6.8 ± 0.9 cells/200 μm) dressings was observed when compared with the control group (5.1 ± 0.6 cells/200 μm, Figure 6F). Neovascularization of wound tissues was examined and showed a significantly higher vessel density in the Ag200 + Mg100 (3.8 ± 0.4%) and Ag2000 + Mg100 (3.9 ± 0.4%) treated wound sections. Moreover, a lesser neovascularization was observed in the control group (1.8 ± 0.3%, *p* < 0.05, Figure 6G,H). From the acute and chronic models, we observed a positive role for Mg in the in vivo wound healing model via a possible action mechanism of increased keratinocyte proliferation and neovascularization.

## 3. Discussion

In this study, we demonstrated that a Mg-Ag-based dressing improved wound healing by increasing fibroblast migration, neovascularization, and basal cell proliferation in both in vitro and in vivo studies. Materials testing showed that Mg and Ag were stably preserved in the constructed material and were able to rapidly release in moist environments. The Mg-based dressing had good cellular compatibility without harmful extractables. Furthermore, Mg enhanced the antibacterial activity of Ag. The present results show that a Mg-Ag-based coating can be manufactured as an optimal dressing for adjuvant wound therapy. 

The influence of Mg in cell physiology has been well-demonstrated. Mg sulfate affected the viability and proliferation of the human gastric adenocarcinoma epithelial cell line in a dose-dependent manner from 3 to 50 mM concentrations [19]. The reduction in osteoclast activity was also reported at a high Mg concentration (>3 mM) [20]. We observed a low expression level of Mg in human wounds. The concept of elevated Mg levels in wound tissue fluids provides intriguing insights into wound healing processes. A related study emphasized that incorporating Mg into hydrogels promoted the differentiation of human skin fibroblasts into myofibroblasts, contributing to skin wound healing [21]. The findings from our study on the diverse distribution of Mg in the body offer valuable insights for developing innovative strategies to improve wound healing outcomes. 

A comprehensive understanding of Mg in wound dressing is essential for optimizing its therapeutic benefits while minimizing potential adverse effects. The potential mechanism by which magnesium-based dressings enhance wound healing is the infiltration of magnesium ions from the dressing into the wound bed. However, further studies are necessary to confirm this proposed mechanism. The release kinetics and biological effects of Mg from wound dressings have not been studied. Our data indicated that the release of Mg peaked on day 3. However, the safety of Mg-containing dressings has raised concerns. Our study on the toxic effects of silver (Ag) and magnesium (Mg) in wound dressings followed a standard cell toxicity test following the ISO 10993-5 guidelines. Two different types of cell lines demonstrated high survival rates when exposed to different concentrations of Ag and Mg from the Mg-Ag-based dressings, indicating the absence of harmful extractables from Ag and Mg in the materials at the concentrations tested. These findings underscore the importance of understanding the cytotoxic effects of wound dressing components like Ag and Mg to ensure their safety and efficacy in promoting wound healing [22,23,24]. The synergistic effect of Ag and Mg on the viability of fibroblasts (NIH-3T3s) observed in our study was intriguing. Further research is warranted to investigate how Ag may enhance the effects of Mg.

The release of Mg and Ag is crucial to determining the bacteriostatic activity of the dressings. It is noteworthy that an effective concentration of silver for bactericidal activity typically ranges from 30 to 40 mg/L [17]. The maximum release potential of Mg (2 mM) and Ag (0.8 ppm) in the Ag200 + Mg100 dressing is sufficient to achieve 40% bacteriostatic activity against *E. coli*, which underscores the synergistic effect of Mg in augmenting the antibacterial efficacy of silver. This innovative concept suggests that the incorporation of magnesium can potentiate the antimicrobial effects of silver, offering a promising strategy for developing wound dressings with enhanced antibacterial properties and advancing the understanding of how different combinations of antimicrobial agents can work synergistically to combat microbial infections effectively. This study’s results provide valuable insights into the development of wound dressings with improved antimicrobial capabilities, paving the way for wound care management and infection control. However, further investigation is needed to compare the efficacy of Mg-Ag dressings with that of commercial Ag dressings.

Our in vitro and in vivo studies demonstrated the promising role of the Mg-Ag-based dressings in enhancing wound healing. The process of wound healing involves various molecular regulators and chemoattractants that promote cell migration. Studies have shown that biosynthesized bacteria nanocellulose/silver nanocomposites can enhance wound healing by accelerating cell migration into the wound area, leading to faster closure of scratches [25]. The presence of Mg has been found to promote fibroblast migration and accelerate wound closure, highlighting its potential therapeutic benefits in enhancing the wound healing process [26]. Our results indicated that Mg might act through mechanisms that increase keratinocyte proliferation and neovascularization. Adequate levels of magnesium are essential for normal keratinocyte proliferation [27]. Magnesium supplementation can boost the expression of hyaluronan synthases, crucial enzymes involved in hyaluronic acid production, thereby improving skin hydration and aiding in wound healing [28]. Mg likely promotes keratinocyte proliferation by stimulating DNA and protein synthesis and modulating cell cycle regulators [29]. Additionally, Mg promotes neovascularization through multiple mechanisms, such as the enhancement of endothelial cell proliferation, migration, and tube formation, upregulating angiogenic genes, reducing oxidative stress, and creating a favorable wound microenvironment [30]. These properties make Mg a promising therapeutic target for improving vascularization in impaired wound healing and osseointegration, especially in the context of diabetes [30]. We developed the Mg-Ag dressing to improve wound healing through skin absorption. Research has demonstrated that magnesium can indeed be absorbed through the skin, offering a potential method for improving magnesium status in the body [31]. Although there is a lack of clinical validation for Mg-Ag dressings, a study conducted on patients with diabetic foot ulcers demonstrated that magnesium supplementation led to a significant improvement in wound healing compared to the placebo [32]. The potential effect of Mg-Ag dressings on chronic wound healing is promising. However, further research is necessary to fully understand the role of magnesium in wound healing. In summary, we have pioneered the utilization of Mg-Ag-based dressings for managing both acute and chronic wounds. Our research highlights the crucial role of magnesium in wound healing. Magnesium contributes significantly to this process by increasing keratinocyte proliferation and neovascularization. Furthermore, the Mg-Ag polyethylene material exhibits superior antibacterial properties, making it a promising candidate for the development of innovative wound dressings. Finally, the dressing’s efficiency was elucidated through in vitro and in vivo studies, which are schematically illustrated in Figure 1.

## 4. Materials and Methods

### 4.1. Patients and Samples

Concentrations of Mg in blood and wound tissue were measured in human participants. Patients were enrolled if they had acute or chronic wounds with obviously devitalized tissue. Exclusion criteria included cases not requiring wound debridement, wounds from malignant cancers, a propensity for bleeding, and pediatric patients. Wound samples were harvested from patients with acute wounds or chronic wounds under the standard wound debridement procedure. Serum for Mg quantification was collected on the same day as the wound tissues were harvested. To determine Mg concentration in wound tissue fluids, we combined the wound tissue samples in equal volumes with radio immunoprecipitation assay (RIPA, Millipore, Sigma, Burlington, MA, USA) buffer to remove unnecessary materials via centrifugation at 4000 rpm for 30 min at 4 °C. The composition of the RIPA buffer included: 50 mM Tris-HCL at pH 8.0, 150 mM NaCl, 1% NP-40 (Igepal, Millipore, Sigma, Burlington, MA, USA), 0.1% SDS, and 0.5% sodium deoxycholate. Mg concentrations in blood and wound fluid samples were examined using the colorimetric endpoint method with the diazonium salt xylidyl blue, using a Roche cobas c 702 machine (Basel, Switzerland). When magnesium reacts with the diazonium salt known as xylidyl blue, it forms a purple complex. This reaction causes a decrease in the absorbance of xylidyl blue, which is directly proportional to the magnesium concentration in the solution. The Mg level was expressed in mg/dL. Values were expressed as the mean ± standard deviation (SD). Informed consent was obtained from all patients participating in this study, and all study procedures, including sample gathering and analysis, were approved by the Institutional Review Board of National Cheng Kung University Hospital (No. A-ER-112-173). 

### 4.2. Mg and Ag Synthesized Materials

Magnesium-ion and silver-ion dressings were produced by KNH Enterprise Co., Ltd., Taipei, Taiwan. In brief, a non-woven polypropylene cloth was ionic sputter coated with pure Mg and/or Ag target on the sputter coater machine for 300 s at different times before scanning electron microscopy (SEM) observation. In the combined Ag and Mg dressing, the silver ions were sputtered first, and then the magnesium ions were sputtered on top of it, so that the magnesium ions are on the outermost layer of the dressing. The thickness of the Mg coating was tested at 25, 50, 100, and 200 nm. The release concentration profile of the 100 nm group was the most ideal, so the following experiment used only the 100 nm as the Mg coating thickness. Two thicknesses of Ag coating, 200 and 2000 mg/m^2^, were fabricated in the pilot study. The groups investigated in this study were the control (polypropylene without ionic coating), Mg100 (only Mg coating at 100 nm), Ag200 + Mg100 (Ag coating at 200 mg/m^2^ and Mg coating at 100 nm), and Ag2000 + Mg100 (Ag coating at 2000 mg/m^2^ and Mg coating at 100 nm) groups.

### 4.3. Immersion Test

A part of the synthesized Mg100 or Ag200 + Mg100 dressing material, approximately 120 cm^2^ (N = 3), was immersed in 20 mL of phosphate buffered saline (PBS) according to the ISO 10993-12:2021 standard [33] for sample preparation and reference materials. The materials were incubated at 37 °C in a 5% CO_2_ incubator for 2, 6, 24, and 72 h, and the concentration of ions contained was measured by inductively coupled plasma mass spectrometry (ICP-MS, Perkin-Elmer-SCIEX, ELAN 6100 DRC II, Woodbridge, ON, Canada). The calibration, matrix matching, and instrument conditions were described as before [34]. Briefly, acetic acid was utilized as a carbon source at the optimum concentration of 10% *v*/*v* for matrix matching in the measurement of Mg and Ag. An ICP-MS multi-element standard solution served for external calibration, while rhodium was employed as an internal standard for all measurements. To prevent particle deposition in the sample introduction system, Triton X-100 (Sigma, Kawasaki-shi, Japan) was added to all solutions at a concentration of 0.1% *v*/*v* as a detergent. Calibration involved a four-point (blank and three standards) approach, executed in a straightforward linear mode. 

### 4.4. Micrographs

The dressing underwent a sputtering process, where Ag and Mg ions were applied using a sputter coater machine for a duration of 300 s. Scanning electron microscopy (SEM) was utilized to examine the surface morphology of the dressing, while the energy dispersive X-ray spectrometer (EDS) was employed to analyze the elemental composition present on the surface of the coated dressing. 

### 4.5. Surface Wettability

The contact angle goniometer (First Ten Angstroms, FTA-1000B, Portsmouth, VA, USA) was used to measure surface wettability. An amount of 5 µL deionized water was dropped on the surface of the dressing, and the surface wettability was measured by the contact angle meter. When the contact angle is lower, the surface is more hydrophilic. If the surface is hydrophobic, the contact angle will be greater than 90°.

### 4.6. Cell Culture

Human umbilical vein endothelial cells (HUVECs, P2 to P8, H-UV001, BCRC, Hsinchu, Taiwan) and mouse fibroblasts (NIH-3T3s, P3 to P5, 60008, BCRC, Taiwan) were used for the cell cytotoxicity and migration assays. HUVECs were cultured in medium containing 10% fetal bovine serum (FBS), 1.5% heparin, 2.5% endothelial cell growth supplement (ECGS), and 1% antibiotic-antimycotic (AA). NIH-3T3s were cultured in medium containing 10% FBS, 100 U/mL penicillin G, 100 µg/mL streptomycin sulfate, 1.25 g/mL amphotericin B, and 1% AA. Both types of cells were cultured at 37 °C with 5% CO_2_ in a humidified atmosphere.

### 4.7. Cytotoxicity Assay

HUVECs and NIH-3T3s (5.0 × 10^3^ cells/well) were plated in 96-well plates and incubated overnight at 37 °C in an incubator with 5% CO_2_. After the cells adhered to the plate, the culture medium was replaced with six different media: one containing 10% dimethyl sulfoxide (DMSO, serving as the positive control), another with 10% FBS (fetal bovine serum, serving as the negative control), and four test media with varying concentrations of Mg and Ag. These were incubated for an additional 24 h. The test media were prepared using extracts from Mg and Ag dressings. Mg100 and Ag200 + Mg100 dressings were incubated with 20 mL of culture medium for 24 h to release Mg and Ag ions. After filtering, four different concentrations of Mg (0.1, 0.5, 1, and 2 mM) and Ag (0.04, 0.2, 0.4, and 0.8 ppm) were diluted with 10% FBS for further experiments. Cell Counting Kit-8 (CCK-8) was used for the determination of the number of viable cells in cytotoxicity assays. An amount of 10 μL of CCK-8 reagent was added per well, the plates were wrapped in aluminum foil and placed in an incubator for 3 h in the dark. Cell viability was determined by measuring the absorbance at 450 nm using an ELISA reader. Cell viability can be calculated using the following formula:(ODtest − ODblank)/(ODnegative − ODblank) × 100% = Cell viability

### 4.8. Antimicrobial Assay

The antimicrobial test was carried out according to ISO 10993-12:2021 and the United States Pharmacopoeia (USP) specifications. The tested materials were immersed in PBS, incubated at 37 °C for 24 h, and then filtered to prepare a material extract. A strain of *Escherichia coli* (PT-FTDH5, Protech, Taipei, Taiwan) was selected for the antimicrobial assay. To prevent the risk of infection and ensure that the microbial load is within an acceptable range, the strain concentration was controlled to be below 100 CFU (colony-forming units) according to the USP (United States Pharmacopeia) specifications to ensure the safety and quality of our products. Extracts of tested materials, such as uncoated dressing, Mg100, Ag200 + Mg100, and commercial Ag dressing (Acticoat, Smith & Nephew, London, UK), were added to each tube along with the same volume of bacteria concentration and incubated for 16 h. An amount of 2 mL of tryptic soy broth (TSB) was also added for bacterial growth. The experimental group was exposed to the same bacterial concentration and the corresponding extract concentration for 16 h. The optical density (OD) at the 600 nm absorption wavelength was measured using an ELISA reader to estimate the bacterial concentration in a liquid [35]. Analytical data can be obtained using the following formula: (Experiment groups − Control group)/Control group × 100 = bacteriostatic activity (%)

### 4.9. Cell Migration Assay

Cell migration experiments were performed using a culture-insert (ibidi GmbH Gräfelfing, Germany). NIH-3T3s (2 × 10^4^ cells/well) were seeded and cultured in a 37 °C incubator containing 5% CO_2_ until the cells’ growth was complete and without gaps. After the cells were confluent, the culture-insert could be removed to leave a 500 μm wide area in the well. The Mg100 and Ag200 + Mg100 dressings were incubated with 20 mL of culture medium for 24 h, then filtered, and 10% FBS was added to create the experimental solutions. The attached cells were fed with test solutions or control medium to observe cell migration. Following saline washes, the wound was photographed by an inverted optical microscope (Olympus, Tokyo, Japan) at the time points of 0, 6, and 24 h.

### 4.10. Animals

Sprague–Dawley rats (male) were purchased from BioLASCO Taiwan Co., Ltd., Taipei, Taiwan and were bred at the National Cheng Kung University animal center. The animal experimental protocol was carried out under the required conditions in accordance with the NCKU animal center standards and approved by the Institutional Animal Care and Use Committee (IACUC approval no: 107266). Diabetic rats were fed a high-fat diet with a 60% fat content. After 4 weeks of feeding, the rats were fasted for 3 days, anesthetized, and subjected to blood sampling from the tail vein (Roche blood glucose meter). Afterward, the rats were injected with streptozotocin 35 mg/kg through the tail vein. Measurement of plasma blood glucose in the fasting state was carried out on the second and sixth days after induction. Diabetes was successfully induced in rats when their fasting blood glucose levels exceeded 250 mg/dL.

### 4.11. Wound Healing Assay

The 5-week-old male SD rats were anesthetized and depilated. Four full-thickness excisional wounds were made on the dorsum of the mice on either side of the midline using a 10 mm biopsy punch. The wound edge was affixed to the underlying tissue using 4-0 nylon for four stitches to prevent skin retraction (Appendix A). Four wounds were randomly covered with uncoated polyethylene, Mg100, Ag200 + Mg100, and Ag2000 + Mg100 dressings, respectively, with the dressings replaced every other day. Each wound was digitally photographed and quantified at days 0, 1, 3, 5, 7, and 9 using ImageJ software 1.48 (National Institute of Health, Bethesda, MD, USA), which was normalized to the original wound areas. The equation for percent wound closure is: Percent wound closure = (1 − wound area/original wound area) × 100%.

### 4.12. Immunohistochemistry and Histological Analysis

On day 9, the animals were sacrificed, and the wound skins were immersed in 10% formalin fixative for 3 days and then subjected to tissue fixation and waxing. The skin tissue was dehydrated and waxed, and then the tissue was embedded into a paraffin block by a paraffin tissue embedding machine. Typically, paraffin wax used in histology for embedding tissue samples has a melting point between 56 and 58 °C. After the tissue sections were cut from the paraffin block using a microtome (4–6 μm) [36], they were stained with hematoxylin and eosin (H&E), Ki67, or CD31 to observe the structural state of the epithelial layer and the dermis layer of the wound healing site. H&E stain was observed through a microscope (Olympus BX 41 and Olympus E-M5II), and the sliced images taken under the objective lens 10× were edited into an entire wound section by Image Composite Editor. A range of 8 to 15 H&E staining images of each group was taken under the microscope objective 4×, the wound panoramic slice was edited by the computer software, and the wound distance and the total length of the epithelialization were measured by the computer software (Image Pro Plus version 7.1). The percentage of re-epithelialization in each group is as follows:(Newly grown epithelial length)/(Wound length) × 100 = re-epithelialization (%) 

The wound pathological sections were analyzed by immunohistochemistry using Ki67 (1:50) to analyze the changes in the number of newborn keratinocytes [37]. The keratinocytes (located in the basal layer) in the epithelialization site were observed under a microscope objective 40× as neonatal keratinocytes. In the quantitative analysis, a total of 3 staining images were taken from different locations in the wound healing area in each group. The position of the basal layer in each epithelial layer of each stain was observed, and the keratinocytes that showed a positive reaction within 200 μm were counted and quantitatively analyzed. Total blood vessel counts of wound skins were analyzed by CD31 (1:100). The number and size of blood vessels composed of vascular endothelial cells in the endothelial layer were observed under a microscope with a 20× objective. In the quantitative analysis, the stained image was measured by Image Pro Plus to measure the area occupied by each group of vascular endothelial cells and the area of the whole image, thereby converting the percentage of blood vessels in the whole image. The CD31 area was calculated by the following equation:(area of blood vessel)/(area of each picture) × 100 = CD31 area (%) 

### 4.13. Statistical Analysis

Statistical analysis was computed using GraphPad Prism 5 statistical software. The absorbance values of the quantitative tests were expressed as the mean ± standard deviation (mean ± SD). Cell invasion area was calculated by Image J and the quantitative tests were expressed as mean ± SD. The wound photos were calculated using the digital image capture system software (Image Pro Plus) and expressed as the mean ± standard error of the mean (mean ± SEM). The data analysis was carried out using a one-way ANOVA. The multiple comparison analysis was followed by Sidak’s post hoc test. When the *p* value is less than 0.05, it indicates a significant difference. 

## Data Availability

The data presented in this study are available on request from the corresponding authors.

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
