# Peer review of "Novel Magnesium- and Silver-Loaded Dressing Promotes Tissue Regeneration in Cutaneous Wounds"

_ijms, 2024, doi:10.3390/ijms25179311_

Round 1

Reviewer 1 Report

Comments and Suggestions for Authors

Section 4. Materials and methods

Subsection 4.1.

1)      Blood and wound tissue Mg concentrations were obtained from human subjects.” Describe inclusion criteria of human subjects included in the study.

2)      “…tissues were centrifuged at 4000 rpm for 30 min at 4 â—¦C to remove unwanted substances” And then?; describe the whole methods to the minimal detail level required to be replicated in a different laboratory.

3)      “…Mg concentration in blood and wound fluid samples were examined using colorimetric endpoint method.…”Describe how wound fluids were collected for analysis. Describe which colorimetric method end point was used.

Subsection 4.2

4)      Describe sputter experimental conditions.

5)      How was the coatings thickness measured.

6)      Two thickness of Ag coating, 200 and 2000 mg/m2 were fabricated in the pilot study. The results of immersion test of both groups did not show a significant difference, so 100 mg/m2 was used for the rest of study. The groups for this study were control (polypropylene without ionic coating), Mg100 (only My coating at 100 nm), Ag200+Mg 100 (Ag coating at 200 mg/m2 and My coating at 100 nm) and Ag2000+Mg100 (Ag coating at 2000 mg/m2 and My coating at 100 nm) were investigated”  Two thickness of Ag coatings were tested, that is 200 and 2000..then why 100 was tested for the rest of the study??. It states that only 200 Ag thickness coating was investigated; then the sample Ag2000/Mg 100 was or was not further tested after immersion test?

Subsection 4.3 immersion test

7) “matrix matching, and instrument conditions was described as before [31]”   manuscript must be self-standing in the essential; minimal experimental details have to be specified always even when referring to previous papers.

Subsection 4.4. Micrographs

8) Improve writing. Dressings were Au or C coated for 300 seconds for SEM observation?? or it means that coatings were Ag or Mg coated for SEM specifically to study samples

Subsection 4.6. cell culture

9) “Human umbilical vein endothelial cells (HUVECs) and fibroblasts (NIH-3T3)” Specify cell line details such as catalog number and company from which cells were acquired. Improve writing to specify HUVECS are human cells while NIH-3T3 are murine cells.

Subsection 4.8.

10) specify origin of bacterial strain used....purchased, isolated from where, etc..?

Specify which commercial silver dressing was used for comparison?

Subsection 4.11

11) “Four wounds were randomly covered with uncoated polyethylene, Mg100, Ag200+Mg100, Ag2000+Mg100 dressings respectively, which was replaced every other day” Why Ag200+Mg100 samples were used on the in vivo model if this samples were chosen to not being further tested in the in vitro experiments?

Section 2 results

Subsection 2.1.

12) “Of interest, we found that only 10% of Mg expression level was observed in wound tissue fluid (0.24±0.01 mg/dL), compared to those in blood serum (2.25±0.21 mg/dL” ….Mg in non-wounded skin tissue should be tested to discard naturally Mg concentrations are higher in skin tissue than in blood. It is confusing, Material and Methods section specifies that wound tissue samples were taken and Mg concentrations measured...were wound tissue samples taken or wound´s fluids samples??

Subsection 2.2.

13) ”at 6 hours to exhibit excellent antibacterial activity…” what is considered as “excellent” antibacterial activity? Dressing only decrease to 20% E coli

Subsection 2.3

14) The results indicated that Mg and Ag ions can be stably preserved in our material by a sputtered process” This does not demonstrate that ions can be stably preserved. It indicates that Ag and Mg was incorporated to the materials and does not indicated if it occurs as ions, part of other compound or in metallic state; and does not indicate whether they are stably preserved or not.

Subsection 2.4

15) “The contact angle greater than 90 degrees is needed to maintain its hydrophobic property to avoid stickiness of the dressing to the tissue wound, reducing secondary damage during dressing change, and also helping to release ions.”  Further support these statements with bibliography.

16) “Additionally, there were no differences of the contact angle between uncoated dressing and all types of Mg dressings (Fig. S3), suggesting the Mg ion do not destroy the material during sputtered process”  This assay does not test whether Mg ion sputtering does not destroy the material during coating, it only demonstrates that after coating the surface still has hydrophobic properties (surface properties; properties of the coating mainly).

17)“. A maximal Mg release property of 2mM was determined to prevent cell toxicity in our design.”  Why was this determined as the non-toxic maximum Mg release? based on?

18) “when culturing in the different levels of Mg concentrations ranging from 0.1 mM to 2 mM (Fig. 2A)” Specify how the different Mg concentration medium were prepared for cell culturing. Improve writing, from methods section it seems that complete extracts were used all the time, as obtained from 20 mL medium to 100 mm samples specimens.

19) “Although decreasing viability of fibroblasts was observed, cell viability could be reversed to higher than 70% in Ag-Mg 114 compounds….” How the synergistic Mg-Ag effect help to the observed increase in cell viability in comparison to Mg or Ag itself?... further try to explain why cell viability increased.

Subsection 2.6

20) “The data showed the Ag200+Mg100 dressing had 40% of bacteriostatic activity of E.coli (Fig. 3B). These results indicated that Mg enhanced antibacterial activity of Ag”  On the contrary: Ag enhances Mg antibacterial activity because dressings with only Ag were not tested, and thus, it cannot be directly compared with commercial Ag dressing unless it has been proved that the amount of silver in commercial dressing and experimental Ag/Mg dressing tested is the same.

Subsection 2.8

21) “3 different concentrations of Mg dressing including 100 mg, 200 mg and 300 mg were developed. The result showed 3 different types of Mg dressings provided the similar wound-closure response (Fig. S4).” Three different Mg concentration are tested in mouse in vivo while only one was determined to be the optimal and studied in vitro.....Explain the bases to use three different groups of dressing in mouse whith no support from in vitro results.

22) “The punch wounds were covered with Mg 100, Ag200+Mg100 and Ag2000+Mg100 dressings respectively”.  Explain the reason behind testing the dressing Ag2000+Mg100 in vivo if no antibacterial or further in vitro testing was performed for this dressing, due to the decision of Ag200+Mg 100 being the optimum

Section 3 Discussion

23) “Furthermore, Mg enhanced the antibacterial activity of Ag”. Same comment for this asseveration as in the results section.

24) “The concept of elevated Mg levels in wound tissue provides intriguing insights into wound healing processes.”  Explain better how the Mg concentration was determined, from wound tissue or from wound exudates? and in case it was from wound tissue how is the level of Mg concentration in healthy skin tissue. If it was determined from wound tissue fluid or exudates, how this relates, or it should be, in comparison to blood plasma? There are previous reports?

25) “The innovative Mg based dressing offers the advantage of promoting wound healing by absorbing Mg from the wound bed.” How was Mg absorption into the dressing from the wound bed studied or explored in the present study?? no studies related to or to corroborate this phenomenon were presented.

26) “Mg-Ag based dressing, indicating the absence of harmful extractables from Ag and” add....at the concentrations tested.

27)“The data showed the Ag200+Mg100 dressing demonstrated 40% bacteriostatic activity against E. coli, highlighting the synergistic effect of Mg in enhancing the antibacterial properties of silver.” This asseveration from the present results and experimental design is right; the opposite asseveration is not right.

28) “Additionally, Mg promotes neovascularization through multiple mechanism, such as enhancement of endothelial cell proliferation, migration, and tube formation, upregulating angiogenic genes, reducing oxidative stress, and creating a favorable wound microenvironment.” Add references to support this.

Comments on the Quality of English Language

Quality of english is ok; nevertheless, the manuscript must be revised to correct some typos such as writing My instead of Mg (magnesium) or suggest-ed or writting the units side by side the numbers 25°C instead of leaving an empty space 25 °C.

Author Response

Our point to point responses to reviewer 1

Section 4. Materials and Methods

  1. “Blood and wound tissue Mg concentrations were obtained from human subjects.” Describe inclusion criteria of human subjects included in the study.

Res 1: Patients were enrolled if they had acute or chronic wounds with obviously devitalized tissue. Exclusion criteria included cases not requiring wound debridement, wounds from malignant cancers, a propensity for bleeding, and pediatric patients. We have added this part in 4.1. Patients and Samples with mark.

  1. “…tissues were centrifuged at 4000 rpm for 30 min at 4 â—¦C to remove unwanted substances” And then?; describe the whole methods to the minimal detail level required to be replicated in a different laboratory.

Res 2: Thanks for your comment. We have revised this part in 4.1. Patients and Samples with mark.

  1. “…Mg concentration in blood and wound fluid samples were examined using colorimetric endpoint method.…”Describe how wound fluids were collected for analysis. Describe which colorimetric method end point was used.

Res 3: The method for wound fluid collection was described in the revised part in 4.1. Patients and Samples. Colorimetric method has become one of the most popular in use for the measurement of Magnesium today. We use the diazonium salt xylidyl blue for colorimetric method end point used. When magnesium reacts with the diazonium salt xylidyl blue, it results in the formation of a purple complex. This process leads to a reduction in the absorbance of xylidyl blue, which correlates directly with the concentration of magnesium in the solution. The magnesium concentration is quantified and reported in mg/dL. This reaction is a common method used in analytical chemistry for determining magnesium levels in various samples. We have added this in 4.1. Patients and Samples with mark. 

  1. Describe sputter experimental conditions.

Res 4: The dressing was sputtered Mg and/or Ag on the sputter coater machine for 300 seconds before Scanning Electron Microscopy (SEM) observation. Sputter experimental conditions for magnesium coating typically involve several key parameters that influence the quality and characteristics of the deposited film. The production of materials is company confidential. We utilize the final product exclusively for various experiments, including anti-bacterial tests, as well as cellular and animal research. We have revised this part in 4.2. Mg and Ag synthesized materials with mark.  

  1. How was the coatings thickness measured.

Res 5: The thickness of the coating was observed by scanning electron microscope (SEM) in our study.

  1. “Two thickness of Ag coating, 200 and 2000 mg/m2 were fabricated in the pilot study. The results of immersion test of both groups did not show a significant difference, so 100 mg/m2 was used for the rest of study. The groups for this study were control (polypropylene without ionic coating), Mg100 (only My coating at 100 nm), Ag200+Mg 100 (Ag coating at 200 mg/m2 and My coating at 100 nm) and Ag2000+Mg100 (Ag coating at 2000 mg/m2 and My coating at 100 nm) were investigated” Two thickness of Ag coatings were tested, that is 200 and 2000..then why 100 was tested for the rest of the study??. It states that only 200 Ag thickness coating was investigated; then the sample Ag2000/Mg 100 was or was not further tested after immersion test?

Res 6: Thanks for your comments. We apologize for any confusion caused by the previous version. We have deleted this sentence and amended some of words in 4.2. Mg and Ag synthesized materials with mark. 

  1. “matrix matching, and instrument conditions was described as before [31]” manuscript must be self-standing in the essential; minimal experimental details have to be specified always even when referring to previous papers.

Res 7: Thanks for your suggestions. We have outlined the procedure succinctly in 4.3. Immersion test with mark.

  1. Improve writing. Dressings were Au or C coated for 300 seconds for SEM observation?? or it means that coatings were Ag or Mg coated for SEM specifically to study samples

Res 8: Thanks for your suggestions. We have improved English writing in 4.4. Micrographs with mark.

  1. “Human umbilical vein endothelial cells (HUVECs) and fibroblasts (NIH-3T3)” Specify cell line details such as catalog number and company from which cells were acquired. Improve writing to specify HUVECS are human cells while NIH-3T3 are murine cells.

Res 9: Thanks for your reminder. We added the details of HUVECs and NIH-3T3 in 4.6. Cell culture with mark.

  1. specify origin of bacterial strain used....purchased, isolated from where, etc..?

Res 10: We have added the details of E-coli in 4.8. Antimicrobial assay with mark.

Specify which commercial silver dressing was used for comparison?

Res.: We have added dressing and company name in 4.8. Antimicrobial assay with mark.

  1. “Four wounds were randomly covered with uncoated polyethylene, Mg100, Ag200+Mg100, Ag2000+Mg100 dressings respectively, which was replaced every other day” Why Ag200+Mg100 samples were used on the in vivo model if this samples were chosen to not being further tested in the in vitro experiments?

Res 11: We have demonstrated the promising effects of the Ag200+Mg100 in in-vitro experiments. However, the specific influence of varying Ag levels on wound healing remains unclear. To explore the effects of increased Ag concentrations on wound healing, we have developed two distinct Ag dressing variants (Ag200+Mg100 and Ag2000+Mg100) for in-vivo studies. We did not observe a statistical significance in the different Ag dressing variants. We have added this part in 2.8. Increased cutaneous wound healing in normal and diabetic mice by Mg dressings with mark.

  1. “Of interest, we found that only 10% of Mg expression level was observed in wound tissue fluid (0.24±0.01 mg/dL), compared to those in blood serum (2.25±0.21 mg/dL” ….Mg in non-wounded skin tissue should be tested to discard naturally Mg concentrations are higher in skin tissue than in blood. It is confusing, Material and Methods section specifies that wound tissue samples were taken and Mg concentrations measured...were wound tissue samples taken or wound´s fluids samples??

Res 12: We apologize for the confusion. Wound fluid was obtained from centrifuged wound tissue samples. The Mg concentration in both blood and wound fluid samples was analyzed and compared. We have amended some of words in new version of manuscript in 4.1. Patients and Samples with mark.

Section 2 Results

  1. ”at 6 hours to exhibit excellent antibacterial activity…” what is considered as “excellent” antibacterial activity? Dressing only decrease to 20% E coli

Res 13: Thanks for your comment. We have revised the word in 2.2. Efficiently released property of Mg and Ag-containing dressing with mark.

  1. The results indicated that Mg and Ag ions can be stably preserved in our material by a sputtered process” This does not demonstrate that ions can be stably preserved. It indicates that Ag and Mg was incorporated to the materials and does not indicated if it occurs as ions, part of other compound or in metallic state; and does not indicate whether they are stably preserved or not.

Res 14: Thanks for your comment. We have revised this part in 2.3. Incorporation of Mg and Ag in wound dressings with mark.

  1. “The contact angle greater than 90 degrees is needed to maintain its hydrophobic property to avoid stickiness of the dressing to the tissue wound, reducing secondary damage during dressing change, and also helping to release ions.” Further support these statements with bibliography.

Res 15: A surface with a contact angle greater than 90 degrees is classified as hydrophobic, which helps to prevent water from wetting the surface. This characteristic is essential for wound dressings to avoid sticking to the tissue, thereby reducing secondary damage during dressing changes and facilitating the release of ions from the dressing material. (Sci Rep 10, 19751, 2020, https://doi.org/10.1038/s41598-020-76471-x). Another study also highlights that hydrophobic surfaces, which exhibit contact angles greater than 90 degrees, are less water-wettable and can effectively protect wounds from moisture, thus minimizing adhesion to the tissue. This property is crucial for maintaining the integrity of the wound and promoting healing. (Advances in Materials Science 19(3):32-45, 2019, DOI: 10.2478/adms-2019-0015).

  1. “Additionally, there were no differences of the contact angle between uncoated dressing and all types of Mg dressings (Fig. S3), suggesting the Mg ion do not destroy the material during sputtered process” This assay does not test whether Mg ion sputtering does not destroy the material during coating, it only demonstrates that after coating the surface still has hydrophobic properties (surface properties; properties of the coating mainly).

Res 16: Your point is right. We have revised the description in 2.4. Good surface wettability of Mg100 and Ag200+Mg100 dressings with mark.

  1. “. A maximal Mg release property of 2mM was determined to prevent cell toxicity in our design.” Why was this determined as the non-toxic maximum Mg release? based on?

Res 17: Based on the report (Ref. 18) indicating that a high Mg status (> 3mM) could lead to reduced osteoclast activity, we have designed the maximal Mg release property of dressing material to be 2mM.

  1. “when culturing in the different levels of Mg concentrations ranging from 0.1 mM to 2 mM (Fig. 2A)” Specify how the different Mg concentration medium were prepared for cell culturing. Improve writing, from methods section it seems that complete extracts were used all the time, as obtained from 20 mL medium to 100 mm samples specimens.

Res 18: Thanks for your suggestions. We have revised this part in 2.5. Free of harmful extractables of Mg100 and Ag200+Mg100 dressings and 4.7. Cytotoxicity assay with mark.

  1. “Although decreasing viability of fibroblasts was observed, cell viability could be reversed to higher than 70% in Ag-Mg 114 compounds….” How the synergistic Mg-Ag effect help to the observed increase in cell viability in comparison to Mg or Ag itself?... further try to explain why cell viability increased.

Res 19: Thank you for your feedback. You raise a good point. We are also interested to explore the synergistic effects of Mg and Ag. However, this was just one of the experiments in our materials study. We have included additional discussion in 3. Discussion with mark.  

  1. “The data showed the Ag200+Mg100 dressing had 40% of bacteriostatic activity of E.coli (Fig. 3B). These results indicated that Mg enhanced antibacterial activity of Ag” On the contrary: Ag enhances Mg antibacterial activity because dressings with only Ag were not tested, and thus, it cannot be directly compared with commercial Ag dressing unless it has been proved that the amount of silver in commercial dressing and experimental Ag/Mg dressing tested is the same.

Res 20: We agree with your point. We have added some discussion in 3. Discussion with mark.

  1. “3 different concentrations of Mg dressing including 100 mg, 200 mg and 300 mg were developed. The result showed 3 different types of Mg dressings provided the similar wound-closure response (Fig. S4).” Three different Mg concentration are tested in mouse in vivo while only one was determined to be the optimal and studied in vitro.....Explain the bases to use three different groups of dressing in mouse whith no support from in vitro results.

Res 21: Thanks for your comment. Initially, the in-vitro and in-vivo studies were conducted concurrently. At that time, we were uncertain which concentration would be optimal for subsequent research. The in-vivo study revealed no statistical significance in the wound healing efficacy of different Mg concentrations. To minimize experimental costs, we opted for 100 mg of magnesium for the in-vitro study.  

  1. “The punch wounds were covered with Mg 100, Ag200+Mg100 and Ag2000+Mg100 dressings respectively”. Explain the reason behind testing the dressing Ag2000+Mg100 in vivo if no antibacterial or further in vitro testing was performed for this dressing, due to the decision of Ag200+Mg 100 being the optimum

Res 22: Even if Ag200+Mg100 was deemed optimal, testing Ag2000+Mg100 could provide insights into whether the increased silver content (Ag2000) offers any additional benefits, such as improved healing outcomes. This exploration could help determine if there are synergistic effects between higher silver concentrations and magnesium. Our in-vivo study showed no statistical significance of Ag concentrations in the wound healing study, which might have influenced the decision to proceed dressing development with optimal Ag concentration.

Section 3 Discussion

23) “Furthermore, Mg enhanced the antibacterial activity of Ag”. Same comment for this asseveration as in the results section.

Res 23: Thanks for your suggestion. He have revised this part in 2.6. Magnesium enhanced antibacterial ability of silver in wound dressing and 3. Discussion with mark.

  1. “The concept of elevated Mg levels in wound tissue provides intriguing insights into wound healing processes.” Explain better how the Mg concentration was determined, from wound tissue or from wound exudates? and in case it was from wound tissue how is the level of Mg concentration in healthy skin tissue. If it was determined from wound tissue fluid or exudates, how this relates, or it should be, in comparison to blood plasma? There are previous reports?

Res 24: The magnesium concentration was determined from wound tissue fluids, which were obtained by centrifuging debrided wound samples. We have updated the term to "wound tissue fluids" in section 3. Discussion, marked to clarify the source of magnesium levels. It was observed that the magnesium concentration in wound tissue fluids is significantly lower than that in blood serum, a finding that has not been previously studied. Additionally, there are no existing studies investigating magnesium levels in local wound tissue fluids or in healthy tissue. The difference in magnesium concentration between wounded and healthy tissue remains unknown. Further research may be necessary to explore these differences.

25) “The innovative Mg based dressing offers the advantage of promoting wound healing by absorbing Mg from the wound bed.” How was Mg absorption into the dressing from the wound bed studied or explored in the present study?? no studies related to or to corroborate this phenomenon were presented.

Res 25: Thanks for your comment. The potential mechanism by which magnesium-based dressings enhance wound healing is the infiltration of magnesium ions from the dressing into the wound bed. However, we lack evidence to confirm this process. We have revised this section in 3. Discussion to reflect our current understanding.  

  1. “Mg-Ag based dressing, indicating the absence of harmful extractables from Ag and” add....at the concentrations tested.

Res 26: Thanks for your suggestion. We have added these words in 3. Discussion with mark.

27)“The data showed the Ag200+Mg100 dressing demonstrated 40% bacteriostatic activity against E. coli, highlighting the synergistic effect of Mg in enhancing the antibacterial properties of silver.” This asseveration from the present results and experimental design is right; the opposite asseveration is not right.

Res 27: We agree with your point. We have revised this section in 3. Discussion with mark.

28) “Additionally, Mg promotes neovascularization through multiple mechanism, such as enhancement of endothelial cell proliferation, migration, and tube formation, upregulating angiogenic genes, reducing oxidative stress, and creating a favorable wound microenvironment.” Add references to support this.

Res 28: Thanks for your suggestion. We have added the reference 28 in 3. Discussion with mark.

Comments on the Quality of English Language

Quality of english is ok; nevertheless, the manuscript must be revised to correct some typos such as writing My instead of Mg (magnesium) or suggest-ed or writting the units side by side the numbers 25°C instead of leaving an empty space 25 °C.

Res.: Thank you for your reminder. We have addressed these errors as indicated.

Please see the attached file for the revised manuscript

Reviewer 2 Report

Comments and Suggestions for Authors

In general, the paper needs a massive English and writing revision. Please, use a professional to get support. The resolution of the images is very poor. Please, improve them. The reviewer cannot read them.

The introduction is too short, and more details should be provided. For instance, why improving the wound healing would be important? preventing infections?

Line 15-16, rephrase the sentence as it is not clear what 10% of Mg level expression means.

Line 33: emphasized is not the correct word.

Line 33-35, rephrase the sentence as it isn't clear.

You should add the following citations for the sentence on the the antimicrobial activity of silver, line 49, to substantiate your claim: 

1) De Mori, A.; Jones, R.S.; Cretella, M.; Cerri, G.; Draheim, R.R.; Barbu, E.; Tozzi, G.; Roldo, M. Evaluation of Antibacterial and Cytotoxicity Properties of Silver Nanowires and Their Composites with Carbon Nanotubes for Biomedical Applications. Int. J. Mol. Sci. 202021, 2303. https://doi.org/10.3390/ijms21072303

2) Luceri A, Francese R, Lembo D, Ferraris M, Balagna C. Silver Nanoparticles: Review of Antiviral Properties, Mechanism of Action and Applications. Microorganisms. 2023;11(3):629. Published 2023 Feb 28. doi:10.3390/microorganisms11030629.

Methods:

The methods are poorly described. For instance, what does the first sentence in 4.1 mean?

Line 317, what are the ''unwanted substances''? Explain with colorimetric assay you have used for Mg quantification and which piece of equipement you have used.

4.3 how did you carry out the release profile? 

Line 334, what is ''My''?

Line 344, rephrase it.

4.5, which instrument did you use to determine the wettability?

4.6, at which passage were the cells used? where did you purchase the cell lines? 2.5% Endothelial --> 2.5% endothelial. keep an eye to the syntax and grammar.

Line 365, you should write 10not 103.

Line 365-367...check the syntax.

It isn't clear if you have used extracts to test the cytotoxicity? be more precise. What does added 10 ul per cell mean? maybe 10 ul of ...were added to each well?

4.8 what do you mean with the strain was below CFU? CFU/ml? give more details?

In general, you must include the supplier for each of the chemicals used?

4.10 I don't think you have chosen a suitable title for this paragraph.

Line 410, what does this sentence mean? be more specific in the paper.

4.12, please give details on the type of paraffin used and the temperature for the embedding procedure. 

LIne 427-428, be more precise, what is a paraffin slicer? do you mean you have used a microtome? which was the thickness of the sections? add a reference for the H&E staining

You have used Ki67 to analyse the keratinocytes...what about the procedure you have followed?the same for CD31.

After you have done all the improvements mentioned above, I will reconsider the manuscript.

Comments on the Quality of English Language

Extensive revision of the English is needed.

Author Response

Our point to point responses to reviewer 2

  1. In general, the paper needs a massive English and writing revision. Please, use a professional to get support. The resolution of the images is very poor. Please, improve them. The reviewer cannot read them.

Res 1: Thanks for your comment. We do our best for writing and figure revision. We have changed Figure 2, 5,and 6 to new ones.

  1. The introduction is too short, and more details should be provided. For instance, why improving the wound healing would be important? preventing infections?

Res 2: Effective wound healing reduces the risk of complications such as infections, which can lead to more severe health issues, including sepsis, amputation, or even mortality. Chronic wounds, if left untreated, can result in significant morbidity and complications that place a burden on healthcare systems and patients alike. We have added this section in 1. Introduction with mark.

  1. Line 15-16, rephrase the sentence as it is not clear what 10% of Mg level expression means.

Res 3:  We have revised the sentence with mark.

  1. Line 33: emphasized is not the correct word.

Res 4: We have corrected the sentence with mark.

  1. Line 33-35, rephrase the sentence as it isn't clear.

Res 5: We have corrected the sentence with mark.

  1. You should add the following citations for the sentence on the the antimicrobial activity of silver, line 49, to substantiate your claim:

  • De Mori, A.; Jones, R.S.; Cretella, M.; Cerri, G.; Draheim, R.R.; Barbu, E.; Tozzi, G.; Roldo, M. Evaluation of Antibacterial and Cytotoxicity Properties of Silver Nanowires and Their Composites with Carbon Nanotubes for Biomedical Applications. Int. J. Mol. Sci. 2020, 21, 2303. https://doi.org/10.3390/ijms21072303

  • Luceri A, Francese R, Lembo D, Ferraris M, Balagna C. Silver Nanoparticles: Review of Antiviral Properties, Mechanism of Action and Applications. Microorganisms. 2023;11(3):629. Published 2023 Feb 28. doi:10.3390/microorganisms11030629.

Res 6: Thank you for your suggestions. We have added these references 1. Introduction with mark.

Methods:

  1. The methods are poorly described. For instance, what does the first sentence in 4.1 mean?

Res 7: Thank you for your suggestion. We have revised this sentence with mark.

  1. Line 317, what are the ''unwanted substances''? Explain with colorimetric assay you have used for Mg quantification and which piece of equipement you have used.

Res 8: We have changed the ''unwanted substances'' to unnecessary materials. The procedure of Mg quantification was revised in 4.1. Patients and Samples with mark. Equipment for Mg quantification was also added.

  1. 3 how did you carry out the release profile?

Res 9: Thank you for your comment. We have revised this part in 4.3.       Immersion test with mark.

  1. Line 334, what is ''My''?

Res 10: We are sorry for the wrong typing. We have corrected it in 4.2. Mg and Ag synthesized materials with mark.

  1. Line 344, rephrase it.

Res 11: We have revised this part in4.3. Immersion test with mark.

  1. 5, which instrument did you use to determine the wettability?

Res 12: The wettability was determined by contact angle goniometer from First Ten Angstroms, FTA-1000B, USA. We have added this machine in 4.5. Surface wettability with mark.

  1. 6, at which passage were the cells used? where did you purchase the cell lines? 2.5% Endothelial --> 2.5% endothelial. keep an eye to the syntax and grammar.

Res 13: Both cell lines are purchased from Bioresource Collection and Research Center in Taiwan. The optimal passage of cells was added in 4.6. Cell culture with mark. We have corrected the wrong words.

  1. Line 365, you should write 103 not 103.

Res 14: Thank you for your reminder. We have corrected the wrong typing in 4.7. Cytotoxicity assay with mark.

  1. Line 365-367...check the syntax.

Res 15: We have revised this section in 4.7. Cytotoxicity assay with mark.

  1. It isn't clear if you have used extracts to test the cytotoxicity? be more precise. What does added 10 ul per cell mean? maybe 10 ul of ...were added to each well?

Res 16: Thank you for your comment. We have updated this section to better explain the steps involved in the 4.7. Cytotoxicity assay with mark.

  1. 8 what do you mean with the strain was below CFU? CFU/ml? give more details?

Res 17: We have revised this section in 4.8. Antimicrobial assay with mark.

  1. In general, you must include the supplier for each of the chemicals used?

Res 18: Thanks for your suggestions. We have added company name for every products.

  1. 10 I don't think you have chosen a suitable title for this paragraph.

Res 19: We have changed the word in 4.10 Animals with mark

  1. Line 410, what does this sentence mean? be more specific in the paper.

Res 20: We have revised this sentence in 4.10 Animals with mark.

  1. 12, please give details on the type of paraffin used and the temperature for the embedding procedure.

Res 21: We have added the details in 4.12. Immunohistochemistry and histological analysis with mark.

  1. Line 427-428, be more precise, what is a paraffin slicer? do you mean you have used a microtome? which was the thickness of the sections? add a reference for the H&E staining

Res 22: Yes, we use microtome. The optimal thickness is 4-6 μm. We have revised this section and also added the reference in 4.12. Immunohistochemistry and histological analysis with mark.

You have used Ki67 to analyse the keratinocytes...what about the procedure you have followed?the same for CD31.

Res 23: Yes, the procedures for using Ki67 and CD31 to analyze keratinocytes are similar in that they both involved immunohistochemistry techniques.

Round 2

Reviewer 2 Report

Comments and Suggestions for Authors

The introduction can still be improved. But the authors have addressed my comments and I am satisfied.

Comments on the Quality of English Language

Minor English revision